# A New Universal Domain Adaptive Method for Diagnosing Unknown Bearing Faults

**DOI:** 10.3390/e23081052

**Published:** 2021-08-16

**Authors:** Zhenhao Yan, Guifang Liu, Jinrui Wang, Huaiqian Bao, Zongzhen Zhang, Xiao Zhang, Baokun Han

**Affiliations:** 1College of Mechanical and Electronic Engineering, Shandong University of Science and Technology, Qingdao 266590, China; 201982050024@sdust.edu.cn (Z.Y.); wangjinrui@sdust.edu.cn (J.W.); bhqian@sdust.edu.cn (H.B.); skd996576@sdust.edu.cn (Z.Z.); 201982050027@sdust.edu.cn (X.Z.); hanbaokun@sdust.edu.cn (B.H.); 2College of Energy and Power Engineering, Nanjing University of Aeronautics and Astronautics, Nanjing 210016, China

**Keywords:** fault diagnosis, rotating machinery, transfer learning, domain adaptation

## Abstract

The domain adaptation problem in transfer learning has received extensive attention in recent years. The existing transfer model for solving domain alignment always assumes that the label space is completely shared between domains. However, this assumption is untrue in the actual industry and limits the application scope of the transfer model. Therefore, a universal domain method is proposed, which not only effectively reduces the problem of network failure caused by unknown fault types in the target domain but also breaks the premise of sharing the label space. The proposed framework takes into account the discrepancy of the fault features shown by different fault types and forms the feature center for fault diagnosis by extracting the features of samples of each fault type. Three optimization functions are added to solve the negative transfer problem when the model solves samples of unknown fault types. This study verifies the performance advantages of the framework for variable speed through experiments of multiple datasets. It can be seen from the experimental results that the proposed method has better fault diagnosis performance than related transfer methods for solving unknown mechanical faults.

## 1. Introduction

Existing deep neural networks have shown superior performance in various diagnostic tasks for rotating component faults due to their impressive feature learning capabilities [1,2,3]. Such networks include the convolutional neural network [4,5], recurrent neural network [6], and restricted Boltzmann machine [7]. The outstanding performances of these networks heavily depend on the pretraining of deep diagnostic networks with real sample data from the same domain as the test data [8]. However, under actual operating conditions, the dataset is often time-varying and unknowable. Improving the generalization capability of a model under variable working conditions has been regarded as a potential solution for solve unknown working conditions.

Domain discrepancy causes the model based on the previous training data to perform poorly with the new test data set [9,10]. The typical solution to this problem is to pre-train the model and fine-tune the diagnostic network trained from the source domain with the feature distribution of the target domain [11], and the method for the marginal distribution alignment of feature spaces is widely used to narrow the distance between two different domains [12]. Li et al. [13] proposed a fault diagnosis model based on multi-scale permutation entropy (MPE) and multi-channel fusion convolutional neural networks (MCFCNN), which constructs a feature vector set by permuting entropy so that the high accuracy and stability of fault diagnosis are realized. Guo et al. [14] reported a new transfer learning network, which gradually realized the multi-module operation of automatic features learning and machine health status recognition through a one-dimensional convolutional network. Singh et al. [15] presented a deep convolution model to diagnose the type of the gearbox fault under the obvious change of speed. The model minimizes the cross-entropy loss of the source domain and the maximum mean discrepancy loss between the two domains to obtain superior diagnostic performance. Hasan et al. [16] proposed a transfer diagnosis framework based on high-order spectral analysis and multitask learning, which can diagnose non-stationary and non-linear rolling bearing signals in combination with different modes of a given fault type. As can be seen from the above-mentioned networks, solving the problem of domain discrepancies has become a tacit prerequisite for current fault diagnosis.

Traditional diagnostic networks usually assume that the label space of the fault samples in the target domain and the source domain is consistent. However, in actual engineering practice, the fault type of the target domain is often difficult to predict, and the fault type label space is often smaller than the source domain fault label space. Therefore, Cao et al. [17] proposed the use of selective weighting to maximize the positive migration of shared tag space data; this approach can achieve the purpose of per-class adversarial distribution matching. Zhang et al. [18] established an importance weighted adversarial network. This network is especially suitable for partial domain adaptation where the number of fault types in the target domain is less than the number of fault types in the source domain, and can effectively reduce the distribution difference to realize knowledge migration and the fault diagnosis of the target sample. Li et al. [19] suggested applying unsupervised prediction consistency schemes and conditional data alignment for partial domain adaptation. This method effectively solves the partial domain adaptation problem that the target domain data under unsupervised training cannot cover the entire healthy label space. Jia et al. [20] proposed a weighted subdomain adaptation network (WSAN), and a weighted local maximum-mean-discrepancy (WLMMD) is introduced to obtain the transferable information and weight of the sample to realize the diagnosis of the fault type. The research on partial domain adaptation pushes the field of intelligent fault diagnosis into a practical setting.

However, only a very small number of networks can cope with the identification and diagnosis of sudden unknown fault types in the existing fault diagnosis models. We cannot know that the fault type of the target samples must belong to the source domain label space when providing unlabeled target samples. Therefore, open set recognition is an urgent problem faced by transfer learning to broaden practical application scenarios. Busto et al. [21] were the first to suggest marking the shared classes of the source and target domains as general classes and constructed an iterative method to solve the labeling problem. Saito et al. [22] modified the description of open set domain adaptation, which allows only the target domain to contain the private label set. His team also added a boundary between the source domain and the target domain to facilitate the separation of unknown fault samples from known fault samples. This method has been widely evaluated in the field. You et al. [23] provided a concept of universal domain in the field of image recognition which allows intersection between source and target domains and provides a benchmark for future related research.

Considering that the current domain transfer methods often assume that the fault type of the test data is the same as the training data set, while ignores that the specific working conditions and label types of the target domain samples are often unpredictable. It is impossible to diagnose the fault type by directly comparing the distribution of the source domain and the target domain. Thus, we propose a new universal domain adaptation (UDA) method for fault diagnosis under the changing conditions of bearing speed. As shown in Figure 1, the model allows different types of faults to exist between data sets and generates a feature center belonging to each fault type for fault diagnosis by learning the fault features of each fault sample. In order to solve the problem of negative model transfer caused by the input of unknown samples into the network, the model proposes three optimization goals, and train the network gradually by optimizing the objective function to alleviate the phenomenon of negative network transfer. The main contributions of this model are as follows:

The proposed model breaks the assumption of the shared label space in the field of mechanical fault diagnosis and proposes the universal domain to solve the fault type samples that did not appear in the training dataset.The proposed network innovatively proposes to rely on source domain samples to generate feature centers of each fault type and determine the fault type based on the distance between the feature extracted from the sample and the feature center.The model introduces Wasserstein distance to measure the marginal probability distribution between different data, and three optimization equations are added to the network training to optimize the model to alleviate the negative transfer problem of the network when solving unknown domains.

In this paper, a new transfer learning model based on universal domain adaptation is proposed and the proposed model is described in detail. The specific article structure is organized as follows. The details of the proposed method for fault diagnosis under changing speed conditions are provided in Section 2. The fault diagnosis experiment with two sets of bearing data is presented in Section 3. Finally, the conclusions are provided in Section 4.

## 2. Research Methods

### 2.1. Proposed Framework

The frame structure of the proposed approach is shown in Figure 2.

The proposed framework adopts two modules, i.e., the feature extractor G and the classifier C. The feature extractor G is composed of 4 fully connected layers, and the dimensions of the samples extracted from each layer are 512, 128, 64, and 16 dimensions. The classifier C is a two-layer Softmax classifier, which is used to diagnose fault sample types. The original source time-domain signal is processed by FFT and input into the feature extractor G to extract the features of the source domain fault sample. The extracted fault features are then classified by the classifier C to extract the feature center of each fault type from the feature signal of the source domain gradually. After the first model pre-classification, the features of the target domain fault samples are added multiple times with tiny noise containing their own features, and the distance from the feature center is measured. The model realizes the fault diagnosis of the target domain samples after many times of learning and training. The training process of the model is described in detail in Section 3.2. The model introduces the following target objects to improve the diagnostic performance and generalization ability.

#### 2.1.1. Classification Loss

Minimizing the classification error of source domain samples is the first optimization goal of the proposed framework. The classifier learns classification knowledge from the labeled samples in the source domain. The standard Softmax regression loss is selected as the objective function [24]. The specific function formula and explanation are as follows:(1)$$L_{C}=-\frac{1}{m}[\sum_{i=1}^{m}\left(1-y^{\left(i\right)}\right)\log\left(1-h_{\theta }\left(x^{\left(i\right)}\right)\right)+\sum_{i=1}^{m}y^{\left(i\right)}\log h_{\theta }\left(x^{\left(i\right)}\right)]$$
where *x*^(*i*)^ and *y*^(*i*)^ represent the input signal of the *i-*th sample and the probability output corresponding to the sample, and *h_θ_*(*x*^(*i*)^) is the set of probabilities of various fault types corresponding to the *i*-th sample.

#### 2.1.2. Feature Loss

Feature loss is used to correct the error loss caused by discarding useless fault type features in the process of extracting feature centers. Feature loss can be expressed as the absolute difference between the feature extracted from the source domain and the feature center generated by the learning process. The function formula [25] is as follows:(2)$$L_{F}(x_{c},o_{c})=\frac{1}{C}\sum_{c=1}^{C}\left|f\left(x_{c}\right)-f\left(o_{c}\right)\right|$$
where *x_c_* and *o_c_* are the *c*-th features of the feature extraction and feature center.

#### 2.1.3. Distance Loss

Wasserstein distance, which is often used to measure the discrepancy between different distributions, can be understood as the minimum consumption under optimal path planning. The Wasserstein distance is used as distance loss to reflect accurately the distance between the two distributions with little to no overlap in the support set with the objective of measuring the overall distance between the feature center and the target domain feature [26]. The function formula is as follows:(3)$$L_{D}(P_{1},P_{2})=\underset{\gamma \in \prod \left(p_{1},p_{2}\right)}{\inf}E_{(x,y)\sim \gamma }[\left\|x-y\right\|]$$
where $\prod \left(p_{1},p_{2}\right)$ is the set of all possible joint distributions that combine the *P*_1_ and *P*_2_ distributions, and *γ* represents the joint distribution of each possible fault type. *x* represents the feature center sample feature and *y* represents the target sample feature.

### 2.2. Training Process

The goal of the model is to identify the fault type of the target domain sample and reduce the domain distance between two identical faults. At the same time, the fault types of unknown samples in the target domain are identified. The training process of the model is shown in the figure below:

Step 1: Figure 3 shows that the network learns the features of the fault types from the source domain samples to form the characteristic centers of multiple fault types. The classifier tries to pre-classify the target domain samples and attempts to shorten the distance discrepancy between domains. Therefore, the source domain classification loss is introduced into the model. The mathematical equation used is as follows:(4)$$L_{C}=-\frac{1}{m}\left[\sum_{i=1}^{m}\sum_{j=1}^{k}I\{y^{\left(i\right)}=j\}\log\frac{\exp(\theta_{j}^{T}\cdot x^{\left(i\right)})}{\sum_{l=1}^{k}\exp(\theta_{l}^{T}\cdot x^{\left(i\right)})}\right]$$
where *m* represents the number of samples in the source domain and I[·] is an index function used to represent the value of the probability that the sample is true. *θ*_1_, *θ*_2_, ..., *θ_k_* ϵ ℜ^*n*+1^ are the parameters of the model and $1/\left(\sum_{j=1}^{k}e^{\theta_{j}^{T}x^{\left(i\right)}}\right)$ normalizes the distribution such that it sums to 1.

Step 2: Tiny noise is mixed into the target domain samples in the classification, and these tiny noises merge the features of the target domain sample extracted from the feature extractor G. The fault features of the target domain samples mixed with the noise will then undergo a slight change. Given that the mixed tiny noise is related to the target sample features themselves, the extracted fault sample feature will be closer to the feature center of its own fault type. The function formula is as follows:(5)$$X_{to}=X_{t}+\lambda \cdot \overset{⌢}{x}\cdot {ο}^{t}$$
where *λ* represents the feature coefficient of the tiny noise. $\overset{⌢}{x}$ is the feature coefficient of the extracted target domain, which is the sample feature extracted from the target domain sample. *o^t^* is Gaussian white noise used as the tiny noise for network training.

Step 3: The network recalculates the distance between the features of the target sample and the center of each fault type after the addition of noise, and the distance loss between the features of the target sample and the feature center is calculated to judge the fault. The specific distance loss function [25] is shown in the following formula:(6)$$L_{dis}=\left|\frac{1}{m^{s}}\sum_{i=1}^{m^{s}}T\left(x_{si}\right)-\frac{1}{m^{t}}\sum_{i=1}^{m^{t}}T\left(x_{ti}\right)\right|$$
where *x_si_* and *x_ti_* are the *i-*th features extracted from the target domain *X_t_* and source domain *X_s_* through the fully connected layer.

The three steps of model training are looped continuously until the expected performance is achieved as shown in Figure 3. The network repeatedly adds tiny noise interference containing the characteristics of the target domain sample to the target domain samples and measures the feature distance to ensure the accurate diagnosis of the fault type of the target domain samples. The stable samples that have been classified accurately do not undergo classification changes after multiple small disturbances are added, whereas the active samples that have been classified incorrectly jump or leave the feature center.

## 3. Experimental Verification

### 3.1. Experimental Dataset Description

The intelligent fault diagnosis methods trained with the labeled data are required to classify the unlabeled data accurately to validate the effectiveness of this method in universal domain transfer learning. Therefore, as discussed in this section, the datasets acquired from two dedicated rotating part workbenches are used for bearing fault diagnosis experiments.

***CWRU*:** The Case Western Reserve University (CWRU) bearing dataset was collected from an experimental platform provided by the CWRU [27]. The CWRU workbench collected sample data of four health conditions at the 6 o’clock position (orthogonal area of applied load) of the deep groove ball bearing on the drive end of the motor housing. The four health conditions were normal condition (NC), inner race failure (IF), outer race failure (OF), and ball failure (BF). The sampling frequency at the time of data collection is set to 48 kHz, and each fault type was run with varying degrees of damage (0.007-, 0.014-, and 0.021-inch fault diameters). Each type of fault data was collected by the test motor running at three different motor speeds (i.e., 1772, 1750, and 1730 rpm) for fault diagnosis. The CWRU dataset information is shown in Table 1.

***SDUST*:** The Shandong University of Science and Technology (SDUST) bearing dataset was collected from a diagnostic test bench specially designed for bearing faults. The time-domain signal of bearings at different speeds of the motor is collected. Figure 4a shows that the bearing fault test bench is composed of a motor, a rotor, a brake, a bearing seat, and two shaft couplings. The cylindrical roller bearing faults in the SDUST dataset contains three single types of fault and a type of composite fault, which are: OF, roller fault (RF), IF, and roller and outer race fault (ROF). Figure 4 is a schematic diagram of the three single types of fault bearings. The collected bearing signals of each type of fault are divided into two fault severity levels: crack 0.2 mm and crack 0.4 mm. The NC time-domain signal was added to the SDUST dataset to obtain nine types of faults as shown in Table 1. Each acceleration sensor was respectively installed in different parts of the bearing seat, and the sampling frequency was set at 25.6 kHz. The motor speed was set to 1500 r/min, 1800 r/min, and 2000 r/min during data collection, and a total of 200 samples were collected for each fault health status at different motor speeds, and each time-domain sample contained 6400 data points.

### 3.2. Compared Methods Description

The proposed method shares the same experimental configuration and test dataset with all the following comparison methods to evaluate the diagnostic advantages of the proposed model.:

**Baseline:** First, a baseline method without a special technique is proposed to diagnose the UDA problem [28]. The feature extractor and classifier are trained under supervision, and the network is used directly for the fault diagnosis of the target domain dataset.

**L1/2-SF:** The L1/2-SF (L1/2 regularized sparse filtering) approach [29] is widely used as an excellent method for bearing and gear fault diagnosis. This method follows the traditional unsupervised machine learning model, thus providing a benchmark for the proposed method.

**WD-MCD:** The Wasserstein Distance—Maximum Classifier Discrepancy (WD–MCD) approach maximizes the output discrepancy of the classifier and combines marginal probability distribution adaptation to focus on the diagnosis of the transfer model [20]. This model is compared with the proposed model because it is a popular transfer learning method.

**BN–SAE:** As a popular method, the Batch Normalization—Stacked AutoEncoders (BNSAE) approach [30] is an adaptive reparametrization algorithm that aims not to optimize but to regularize the model. The effect of the healthy data classification scheme can thus be examined.

### 3.3. Experimental Results Display

As reported in this section, the UDA problem is experimentally verified. The model parameters during the experiment are shown in Table 2 and are mainly determined in accordance with the verification results of the diagnostic task.

The detailed task information is shown in Table 3. This information randomly selects the fault type for fault diagnosis in the two datasets. In order to verify the fault diagnosis performance of the framework under variable speed conditions, the experiment sets the fault type diagnosis in the phase of gradually increasing bearing speed and the fault type diagnosis in the phase of gradually decreasing speed. The “source classes” represent the fault type of the source domain training sample, the type of fault marked red represents the private type of the source domain (the category does not appear in the target domain fault sample), and the “unknown class” represents the fault type that has not been learned during source domain training. A total of 100 labeled samples under each machine condition are randomly selected as source domain data for model training, and 100 unlabeled samples are used as target domain samples for experimental verification. An average of 10 experiments for each group of results is performed to reduce the effect of randomness.

#### 3.3.1. CWRU Task Set Result Analysis

Table 4 shows the fault diagnosis results of the proposed model using the CWRU dataset for different universal domain tasks. The selected comparison approaches are currently the highly popular domain adaptive and transfer learning methods. The stable diagnostic performance presented by the proposed method in the CWRU dataset task group shows its superiority in solving universal domain adaptation. Furthermore, the proposed method generally obtains smaller standard deviations than other models when performing different tasks, indicating that it has good convergence in the experimental process.

The L1/2-SF, the WD-MCD, and the DA-BNSAE methods achieve relatively ideal accuracy rates in the tasks with the CWRU dataset compared with the baseline method, and their accuracy in some tasks is as high as 80 or more. These methods generally have good feature recognition capabilities in the diagnosis of minor faults. However, in serious fault diagnosis, the proposed model has superior feature recognition and diagnosis performance. Typical task cases are represented by tasks A3 and A7. The accuracy of the proposed method is as high as over 90%, and high accuracy is also obtained in the process of solving task 9. Task 4 is specially set as the task benchmark, which has no unknown fault type and belongs to pure rotation speed transfer. It can be seen that the proposed method can still guarantee high diagnostic accuracy. Comparing the standard deviation of each method reveals that the proposed method exhibits relatively stable performance in multiple tasks, further verifying its convergence performance. Although the performance in minor fault diagnosis shown by the proposed method is not as good as that of the three comparison methods, the superiority of the modified model can be seen in the overall task performance comparison. Moreover, the proposed model is more suitable for fault diagnosis problems in the universal domain than other comparison methods.

The t-distributed Stochastic Neighbor Embedding (t-SNE) method [31] is widely used in the display of various fault diagnosis results. This approach can reduce the dimensionality of the output of the high-dimensional features by the model and provide visualization processing. The A1 task based on the CWRU dataset is selected as the demonstration experiment for t-SNE dimensionality reduction processing for visually displaying the fault diagnosis performance of each network as shown in Figure 5.

The baseline approach exhibits poor fault diagnosis performance as illustrated in Figure 5a. It not only fails to aggregate various types of faults completely, but it also clusters BF14 faults and BF7 faults together. As can be seen in Figure 5b,d, the L1/2-SF, and the DA-BNSAE approach incorrectly classify the BF14 fault as the BF7 fault, and the L1/2-SF approach also shows that the target domain and the source domain samples of the OF7 fault are not clustered. As presented in Figure 5c, although the WD-MCD approach has a better fault clustering effect than the previous three methods, a situation wherein the fault type BF14 is mistakenly classified as a healthy sample exists. The clustering dimensionality reduction graph of the proposed method is shown in Figure 5e. Although a small number of IF14 fault samples are close to the BF14 fault in the proposed method, the target domain and the source domain samples of various types of faults have obvious domain boundaries and show a good fault classification effect.

Figure 6 depicts that the training accuracy and testing accuracy of the method tends to stabilize as the training progresses, and feature and distance losses in the model gradually decrease as accuracy increases.

#### 3.3.2. SDUST Task Set Result Analysis

The experimental accuracy for the SDUST dataset is shown in Table 5. Given that this dataset has more drastic speed changes, diagnostic performance with this dataset is worse than that with the CWRU dataset. It can be seen that the three comparison methods show good diagnostic performance in individual tasks. However, the proposed method is more convincing in terms of overall fault classification effect and stability. The model still provides superior fault diagnosis accuracy under variable speed conditions that further validates its robustness and superiority for UDA problems.

Considering that the experimental task is too heavy, the B1 task based on the SDUST dataset is selected as the demonstration experiment for t-SNE dimensionality reduction processing for visually displaying the fault diagnosis performance of each network (Figure 7). Although the baseline approach provides a good clustering of the source domain fault types, it shows a small amount of confusion between the fault type 3 (NC) and the fault type 8 (ROF0.2), as well as between the fault type 8 (ROF0.2) and the fault type 1 (IF0.2) in the target domain. The baseline approach mistakenly classifies the target domain fault type 8 (ROF0.2) sample as the fault RF0.2 as shown in Figure 7a. As can be seen from Figure 7c, the WD-MCD approach incorrectly classifies the NC samples at 1800 speed as the OF0.2 fault, and some samples as the fault type 8 (ROF0.2) in the target domain are mixed with the fault RF0.2. The DA-BNSAE network, one of the approaches used for comparison, confuses the fault type boundaries of IF0.2, RF0.2, and ROF0.2 faults as presented in Figure 7d. Comparing the proposed method with the L1/2-SF approach, it is found that although the L1/2-SF method has a good clustering effect on the source domain samples, there are still a small number of OF0.2, RF0.2, and ROF0.2 fault samples that are misclassified. The proposed network not only has a more obvious clustering effect on samples of various fault types but also has obvious separation between samples of different fault types, as shown in Figure 7b,e.

Therefore, the proposed model can diagnose the samples of the unknown fault types more effectively than other networks.

## 4. Conclusions

This paper presents a new UDA method for bearing fault diagnosis under different working conditions, which breaks the assumption that the traditional domain adaptive network shares the label space and attempts to solve the unknown scale domain by using a universal label domain method. The proposed method was compared with the current popular domain adaptation methods in the experimental verification stage. Under the premise of sharing the experimental configuration and dataset, we set up multiple sets of experimental tasks for different actual work needs. Through multiple experimental verifications, it is concluded that the proposed method has higher classification accuracy and robustness than the comparison methods in diagnosing bearing datasets under variable conditions, and it can still guarantee high diagnostic performance even in the presence of bearing samples of unknown fault types. Therefore, the proposed method is more suitable for actual working conditions that change from time to time.

## Figures and Tables

**Figure 1 entropy-23-01052-f001:**
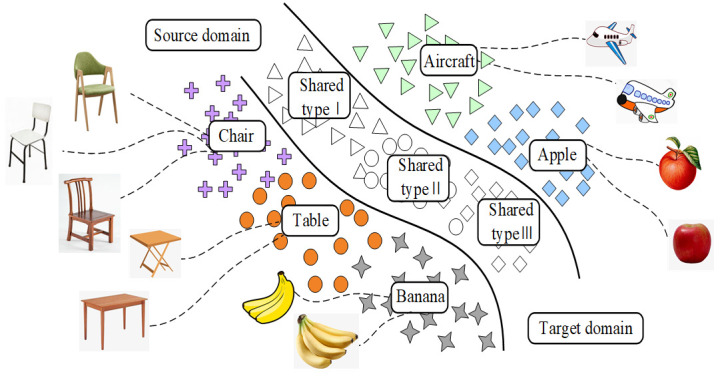
Universal domain adaptation setting (unshaded shapes indicate shared labels).

**Figure 2 entropy-23-01052-f002:**
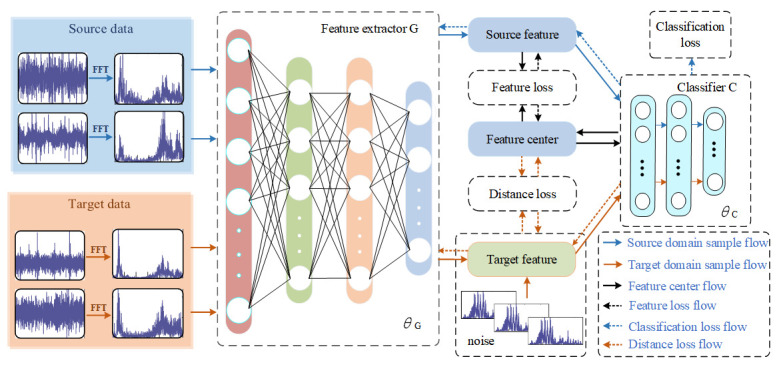
Framework diagram of the proposed method.

**Figure 3 entropy-23-01052-f003:**
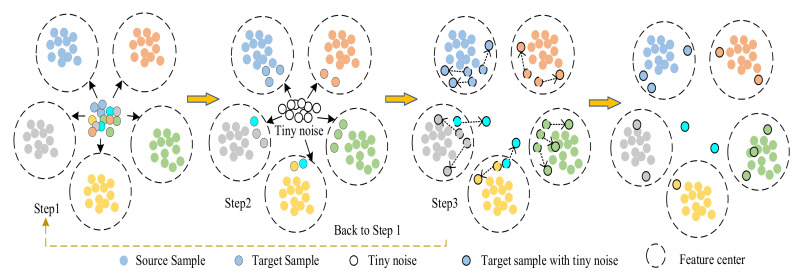
Model training steps.

**Figure 4 entropy-23-01052-f004:**
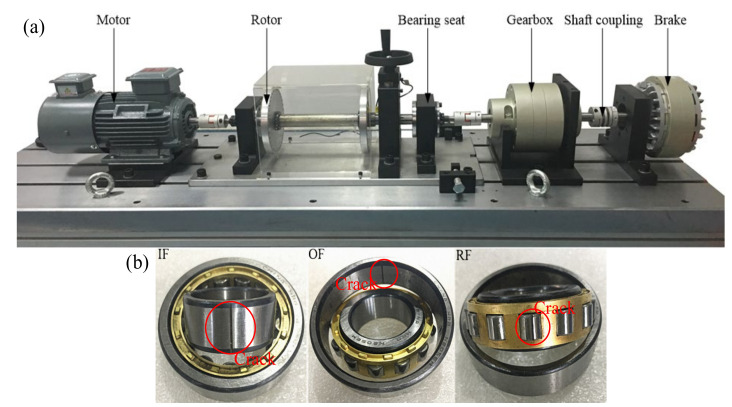
(**a**) Bearing fault test rig and (**b**) three single types of fault bearings.

**Figure 5 entropy-23-01052-f005:**
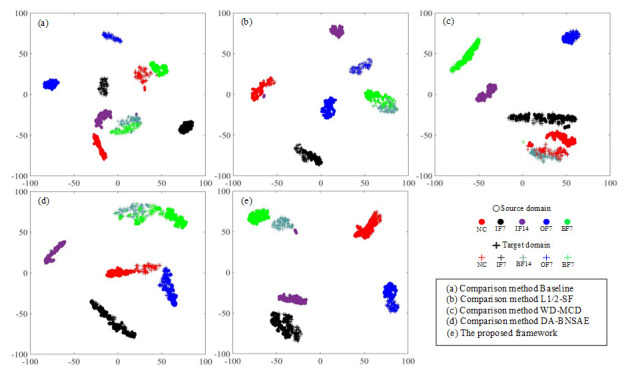
Feature visualization of the t-SNE results for the CWRU dataset in the A1 task.

**Figure 6 entropy-23-01052-f006:**
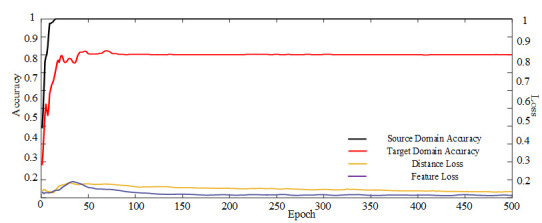
Accuracy and training loss of the proposed method in the A1 task.

**Figure 7 entropy-23-01052-f007:**
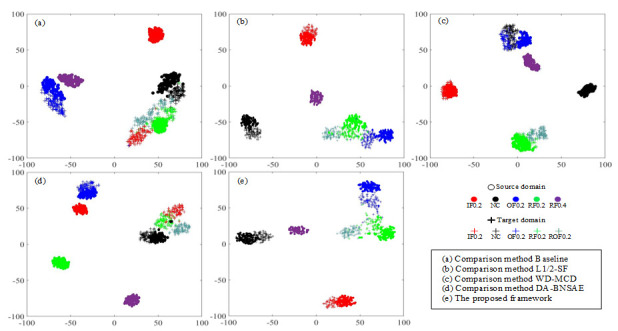
Feature visualization of the t-SNE results for the SDUST dataset in the B1 task.

**Table 1 entropy-23-01052-t001:** Information of the two datasets.

Dataset	Class Label	1	2	3	4	5	6	7	8	9	10
CWRU	Fault location	N/A	IF	IF	IF	BF	BF	BF	OF	OF	OF
	Fault size (mil)	0	7	14	21	7	14	21	7	14	21
SDUST	Fault location	IF	IF	N/A	OF	OF	RF	RF	ROF	ROF	
	Fault size (mm)	0.2	0.4	0	0.2	0.4	0.2	0.4	0.2	0.4	

**Table 2 entropy-23-01052-t002:** Parameters used in this study.

Parameter	Value	Parameter	Value
Epochs in general training *E*_s_	2000	Sample dimension *S*	1200
Epochs in testing *E*_t_	500	Feature center dimension *C*_d_	16
Batch size *B*_s_	10	First-level feature dimension *f*_1_	512
Dropout_rate *D*	0.1	Second-level feature dimension *f*_2_	128
Learning rate *L*r	0.001	the feature coefficient of the tiny noise *λ*	0.05

**Table 3 entropy-23-01052-t003:** Universal domain transfer learning task information.

CWRU				SDUST			
Task Name	Transfer (Speed)	Source Classes	Unknown Class	Task Name	Transfer (Speed)	Source Classes	Unknown Class
**A1**	1730 → 1750	1, 2, 3, 5, 8	6	**B1**	1500 → 1800	1, 3, 4, 6, 7	8
**A2**	1730 → 1750	1, 2, 3, 6, 9	8	**B2**	1500 → 1800	2, 3, 4, 5, 7	9
**A3**	1730 → 1750	1, 4, 5, 7, 10	2	**B3**	1500 → 1800	1, 3, 5, 6, 8	7
**A4**	1730 → 1750	No unknown fault		**B4**	1500 → 1800	No unknown fault	
**A5**	1730 → 1772	1, 2, 3, 5, 8	6	**B5**	1500 → 2000	1, 3, 4, 6, 7	8
**A6**	1730 → 1772	2, 3, 5, 7, 9	1	**B6**	1500 → 2000	1, 3, 4, 5, 6	8
**A7**	1730 → 1772	4, 5, 6, 8, 9	3	**B7**	1500 → 2000	1, 3, 5, 6, 8	2
**A8**	1750 → 1730	1, 2, 3, 5, 8	6	**B8**	1800 → 1500	1, 3, 4, 6, 7	8
**A9**	1750 → 1730	1, 4, 5, 7, 10	2	**B9**	1800 → 1500	1, 2, 3, 4, 8	6
**A10**	1772 → 1730	1, 2, 4, 5, 8	10	**B10**	2000 → 1500	1, 3, 4, 6, 8	9

**Table 4 entropy-23-01052-t004:** Means of the testing accuracies in different tasks with the CWRU dataset (%).

Method	Baseline	L1/2-SF	WD-MCD	DA-BNSAE	Proposed
**A1**	54.67 (±12.4)	82.2 (±6.3)	80.08 (±2.6)	78.23 (±0.4)	82.98 (±2.1)
**A2**	43.2 (±8.4)	73.68 (±5.1)	75.02 (±3.2)	74.22 (±2.1)	80.32 (±1.4)
**A3**	57.46 (±3.7)	70.31 (±8.2)	82.38 (±5.0)	73.65 (±3.4)	90.31 (±2.7)
**A4**	72.56 (±6.2)	98.83 (±1.7)	99.43 (±0.5)	98.53 (±1.6)	99.88 (±0.2)
**A5**	55.07 (±4.6)	69.45 (±1.2)	80.2 (±0.4)	71.07 (±2.7)	80.96 (±6.5)
**A6**	47.9 (±3.0)	69.67 (±3.2)	72.15 (±3.8)	65.2 (±2.3)	77.38 (±9.3)
**A7**	33.3 (±6.4)	64.78 (±11.7)	73.06 (±5.4)	64.93 (±15.1)	92.37 (±2.2)
**A8**	48.47 (±10.7)	74.32 (±3.6)	79.6 (±3.3)	75.54 (±6.5)	84.31 (±3.8)
**A9**	48.78 (±7.1)	71.56 (±5.9)	70.34 (±8.4)	67.3 (±7.9)	88.01 (±4.5)
**A10**	44.7 (±15.3)	55.62 (±5.5)	71.62 (±7.8)	65.07 (±13.6)	85.96 (±7.4)
**Average**	50.61	73.04	78.39	73.37	86.25

**Table 5 entropy-23-01052-t005:** Means of the testing accuracies in different tasks with the SDUST dataset (%).

Task	Baseline	L1/2-SF	WD-MCD	DA-BNSAE	Proposed
**B1**	44.8 (±3.4)	79.44 (±4.6)	77.32 (±2.7)	77.18 (±3.9)	80.04 (±2.1)
**B2**	41.45 (±7.5)	72.72 (±2.4)	67.26 (±5.6)	68.67 (±7.2)	83.06 (±5.7)
**B3**	47.67 (±4.3)	71.32 (±8.5)	76.84 (±3.0)	76.57 (±2.9)	81.96 (±4.3)
**B4**	70.42 (±3.4)	97.53 (±1.4)	99.44 (±0.9)	98.94 (±1.1)	99.76 (±0.5)
**B5**	45.07 (±6.3)	75.97 (±3.2)	78.67 (±9.6)	64.87 (±4.1)	84.72 (±5.9)
**B6**	46.3 (±7.2)	67.05 (±10.5)	76.1 (±13.7)	58.31 (±17.3)	79.9 (±3.5)
**B7**	52.78 (±3.8)	72.14 (±7.2)	76.18 (±9.4)	72.23 (±4.5)	90.7 (±2.9)
**B8**	51.21 (±5.2)	78.3 (±5.9)	81.04 (±2.1)	64.01 (±7.9)	90.44 (±4.7)
**B9**	44.23 (±6.5)	49.52 (±17.8)	64.4 (±14.5)	60.03 (±15.9)	82.64 (±7.9)
**B10**	39.78 (±6.3)	57.23 (±5.6)	45.63 (±6.7)	57.22 (±9.0)	73.62 (±12.3)
**Average**	48.37	72.12	74.29	69.80	84.68

## Data Availability

The data used to support the findings of this study are available from the corresponding author upon request.

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
