# Peer review of "A New Universal Domain Adaptive Method for Diagnosing Unknown Bearing Faults"

_entropy, 2021, doi:10.3390/e23081052_

Round 1

Reviewer 1 Report

Dear Authors,

Major remarks:

  1. Section 2.1 and 2.2 should be considered as part of the introduction.
  2. The literature review should be revised and should take into account the whole world literature
  3. Please provide more information about training and values of the parameters used during training of the model (i.e. the value of the tiny noise parameter),
  4. The whole paper should be revised in the context of the missing “a/an/the”,
  5. Conclusions should be revised and give more concluding information (at the moment it looks like an abstract of the paper).

Minor remarks:

  1. Line 91-92 – „…marginal probability distribution distribution between different…” grammar
  2. Lines 95-99 – the numbers should be in the same form as the numbering of the sections.
  3. Lines 132-133 - “In this section, a new transfer learning model based on universal domain adaptation is proposed and the proposed model is described in detail.” – This information should be given in the description of the paper (lines 95-99).
  4. Line 135 “structrue” – typo.
  5. Line 239 “diffesent” – typo
  6. Line 247-248 “This method and the comparison method share same dataset for testing to evaluate the advantages of the proposed method.” – grammar/style, please revise
  7. Table 4 should be given after citation in the text

Best Regards.

Reviewer 2 Report

The authors proposed an universal domain adaptive bearing fault diagnosing method. Suggestions for the authors are as follows

About writing

1) The sentence 'Therefore, a new universal domain 16 adaptive network ... network to fail to classify is proposed,' is better to transform to 'Therefore, a new universal domain 16 adaptive network has been proposed aiming at ... network failed to classify,'.

2) The word 'distribution' in line 92 occurs two times and one of them should be deleted.

3) If the formula (2) is not defined by the authors, the reference should be cited.

4) 'LD' is used in both formula (3) and (6). It would be better to change one of them.

About content

1) It would be better to combine section 1 the introduction and section 2 the related work. Because in section 2, nothing important has shown.

2) Why using the frequency domain spectrum but not the original time domain signal is used to extract the features?

3) Please explain what is the meaning of 'x' and 'y' in formula (3) in your manuscript.

Round 2

Reviewer 1 Report

Dear Authors,

Please check grammar/style in the following lines:

  • 20-22 "a new universal domain adaptive network has been proposed aiming at the problem that the unknown fault type in the target domain may cause the traditional transfer learning network to fail to classify"
  • 417 "Although the three comparison methods"
  • 420 " speed change scenarios "

Kind Regards,

Author Response

请看附件

Reviewer 2 Report

The authors have revised the manuscript carefully. The manuscript can be published at the present form.

Author Response

We are very grateful to the reviewers for their review and affirmation of the article, and we will continue to work hard for our future work.